# Phylogenetic investigation and mitochondrial genome description of ten species in nine genera of Cicadellinae from China (Hemiptera: Cicadellidae)

Likun Zhong[1,2], Yan Jiang[1,2], Xiaofei Yu[3], Bin Yan[1,2], Renhuai Dai[1,2], Maofa Yang[1,2,3]*

**1** Institute of Entomology, Guizhou University, Guiyang, China, **2** Guizhou Provincial Key Laboratory for Agricultural Pest Management of the Mountainous Region, Guiyang, China, **3** College of Tobacco Sciences, Guizhou University, Guiyang, China

* gdgdly@126.com

## Abstract

The Cicadellinae subfamily of the Cicadellidae (Hemiptera: Auchenorrhyncha: Cicadelloidea) is quite large, with about 2400 species across 330 genera worldwide, of which 263 species in 23 genera occur in China. This work involved the sequencing of the mitochondrial genomes of ten species and nine genera within the Cicadellinae family: *Anagonalia emeiensis*, *Anagonalia melichari, Anatkina vespertinula*, *Erragonalia choui*, *Gunungidia aurantiifasciata*, *Kolla paulula*, *Nanatka castenea*, *Paratkina nigrifasciana*, *Seasogonia rosea*, and *Stenatkina angustata.* The acquired mitogenomes displays a significant AT bias, with the AT contents ranging from 76.1% to 81.7%. The lengths of the mitochondrial genomes range from 14,768 bp to 16,194 bp. TAG and a single T are less frequently used as stop codons, whereas TAA is the most frequently used one. Less PCGs begin with T/GTG, and the majority begin with the conventional ATN (ATA/T/G/C) codon. All tRNA genes in Cicadellinae mitogenomes fold into the standard secondary structure of a cloverleaf,except for trnS1, which lacks a stable dihydrouridine (DHU) stem and instead has s simple loop. Six phylogenetic trees constructed by maximum likelihood (ML) and Bayesian inference (BI) for the three mitochondrial datasets, respectively, consistently showed intergeneric and interspecific relationships within the Cicadellinae. Our results in particular shed light on the molecular and phylogenetic evolution of the pronotum in Cicadellinae.

## Introduction

The family Cicadellidae (Hemiptera: Auchenorrhyncha: Cicadelloidea) has the comparatively large subfamily Cicadellinae, which is found all over the world. Approximately 2400 species in 330 genera have been identified globally; 263 species in

**Data availability statement:** All relevant data are within the manuscript and its Supporting Information files.

**Funding:** This research was supported by the Guizhou Province Science and Technology Innovation Talent Team Project (grant No. Qian Ke He Pingtai Rencai–CXTD [2021]004), and the National Natural Science Foundation of China (32360125).

**Competing interests:** The authors have declared that no competing interests exist.

23 genera occur in China [1–6]. The insects of this subfamily are medium to large in size, 4–19 mm in length, and usually cylindrical. The ocelli are located in the center of the crown or near the base; the lateral frontal suture extends up to the crown or close to the ocelli; the anterior lateral lamellae of the prothorax are exposed; and the tibial setae of the hind feet are regularly arranged in four rows. The species are widely distributed throughout the world's major zoogeographic regions, especially in the tropics and subtropics, where they are most abundant. Some of the leafhoppers are also pests that harm the agricultural economy by stinging the plant's sap and leaving wounds on the plant's surface that are susceptible to infections such as viruses or bacteria. [7–10]. For pest control, accurate species identification is crucial. Some taxa that are based on morphology have a contentious taxonomic position, nevertheless. Consequently, phylogenetic analysis and insect identification are thought to benefit greatly from the use of molecular data. The majority of Cicadellinae researchers mostly work on molecular fragment phylogenetic studies and traditional classification [11–14]. The mitochondrial genomes of the subfamily Cicadellinae have been reported for the genera *Atkinsoniella*, *Bothrogonia*, *Cicadella*, and *Cofana*, mostly focusing on phylogenetic studies within a particular genus [15–20].

The mitogenome, a double-stranded molecule of approximately 14–17 kb in size that typically contains 13 protein-coding genes (PCGs), 22 transfer RNAs (tRNAs), 2 ribosomal RNAs (rRNAs), and a noncoding A+T-rich region, is the most extensively studied genomic system in insects [21–23]. Because of its maternal inheritance, quick rate of evolution, and significantly more conserved gene content than nuclear genes, it has been extensively employed as a source of sequence data for phylogenetic analysis in the last few decades to study insect taxonomy, phylogenetic relationships, evolution, and biogeography [24–27]. The gene organization of the leafhopper mitogenome is conserved [28–30]. In order to provide new information and genomics data for phylogenetic research on Cicadellinae, we reconstructed the phylogenetic relationships of the family Cicadellidae using maximum likelihood and Bayesian inference methods, combined with other available sequence data in GenBank.

## Materials and methods

### Taxon sampling, DNA extraction, and sequencing

S1 Table in S1 File displays the specifics of the ten specimen collections. Each collection was stored at the Institute of Entomology, Guizhou University, Guiyang, China (GUGC), labeled with comprehensive collecting information, and preserved in 100% ethanol. Species-level morphological identification was carried out according to the description of Yang *et al*. [4]. The DNeasy® Tissue Kit (Qiagen, Hilden, Germany) was used to extract the whole DNA from the head and thoracic muscle tissue of each specimen in compliance with the manufacturer's instructions. The remaining abdomen and wings are preserved by number at the Institute of Entomology, Guizhou University. After preparing the libraries, whole genomic DNA was sequenced at Berry Genomics (Beijing, China) utilizing the Illumina NovaSeq6000 platform with 150 bp paired-end reads. S2 Table in S1 File with summary statistics of the sequenced species and raw reads have been deposited in the public repository (SRA) and made available (S3 Table in S1 File).

### Sequence assembly, annotation, and analysis

Using NOVOPlasty 2.7.2 [31] or GetOrganelle 1.7.4.1 [32], about 5 Gb of clean data for each species were assembled. We used the MITOS web server (http://mitos.bioinf.uni-leipzig.de/index.py) and BLAST searches in NCBI (https://blast.ncbi.nlm.nih.gov/Blast.cgi) to annotate the assembled sequences with invertebrate genetic codes [33] and the locations and secondary structure predictions of 22 common tRNAs using the search service tRNAscan 1.21 [34]. We were able to identify rrnL and rrnS based on the gene positions of nearby tRNAs and similarities with other leafhopper mitogenomes in NCBI. Thirteen PCG sites containing invertebrate genetic codes were predicted using Geneious Prime's Open Reading Frame Finder. The CGView Server (http://stothard.afns.ualberta.ca/cgview server) [35] and Adobe Illustrator CS6 were used to create the mitogenome maps. Utilizing MEGA 7.0 software, PCGs' base composition and codon use were determined [36]. The following formulas were used to compute the chain asymmetry: Both AT and GC skews are equal to $(A - T)/(A + T)$ and $(G - C)/(G + C)$ [37]. Utilizing TBtools v2.010 (Beijing, China), a heatmap representing PCGs' relative synonymous codon usage (RSCU) values was produced [38]. Thirteen aligned PCGs were utilized to determine the rates of synonymous substitutions (Ks) and nonsynonymous substitutions (Ka) using DnaSP v6.12.03. [39].

### Phylogenetic analyses

With the support of many plug-in applications, PhyloSuite [40] was used to conduct, manage, and streamline the analyses. The nucleotide sequences of 13 protein-coding genes (PCGs) and two ribosome rRNA (12S rRNA + 16S rRNA) and amino acid (13 PCGs) sequences were aligned in batches with MAFFT [41] using the '--auto' strategy and codon alignment mode. With the use of MACSE v. 2.03 [42], a codon-aware technique that maintains the reading frame and permits the insertion of sequencing with frameshifts or errors in sequencing, the alignments were refined. Using Gblocks [43] and the following parameter settings, ambiguously aligned fragments of 13 alignments were eliminated in batches: with the following parameter settings: minimum number of sequences for a conserved/flank position (36/36), maximum number of contiguous nonconserved positions (8), minimum length of a block (10), and allowed gap positions (with half). The best partitioning scheme and evolutionary models for 13 predefined partitions were selected using PartitionFinder2 [44], with a greedy algorithm and AICc criterion. Maximum likelihood phylogenies were inferred using IQ-TREE [45] for 20000 ultrafast [46] bootstraps, as well as the Shimodaira–Hasegawa–like approximate likelihood-ratio test [47]. Using MrBayes 3.2.6 [48] and a partition model (10000000 generations), Bayesian inference phylogenies were constructed, with the first 25% of sampling data being removed as burn-in. The iTOL online tool was used to browse and edit ML and BI trees (https://itol.embl.de/) [49].

## Results

### Genome structure and nucleotide composition

Ten recently sequenced species of Cicadellinae have had their whole mitochondrial genomes acquired and added to the GenBank database: *Anagonalia emeiensis* (GenBank: NC073096; length: 14,768 bp), *Anagonalia melichari* (GenBank: MT642611; length: 15,398 bp), *Anatkina vespertinula* (GenBank: NC070007; length: 15,559 bp), *Erragonalia choui* (GenBank: NC070001; length: 16,503 bp), *Gunungidia aurantiifasciata* (GenBank: MT622821; length: 15,669 bp), *Kolla paulula* (GenBank: MW542170; length: 14,995 bp), *Nanatka castenea* (GenBank: NC070000; length: 14,953 bp), *Paratkina nigrifasciana* (GenBank: NC069999; length: 16,194 bp), *Seasogonia rosea* (GenBank: NC070006; length: 15,293 bp), and *Stenatkina angustata* (GenBank: NC069998; length: 15,173 bp). *E choui* and *A emeiensis* had the longest and shortest mitogenomes, respectively. All ten mitogenomes displayed the same gene arrangement and orientation (Fig 1). All mitogenomes had a control region along with 37 mitochondrial genes (22 tRNA genes, two rRNA genes, and 13 PCGs). The major strand of the J chain included 23 genes (9 PCGs and 14 tRNAs), while the minor strand of the N chain contained 14 genes (4 PCGs, 8 tRNAs, and 2 rRNAs) (S4 Table in S1 File).

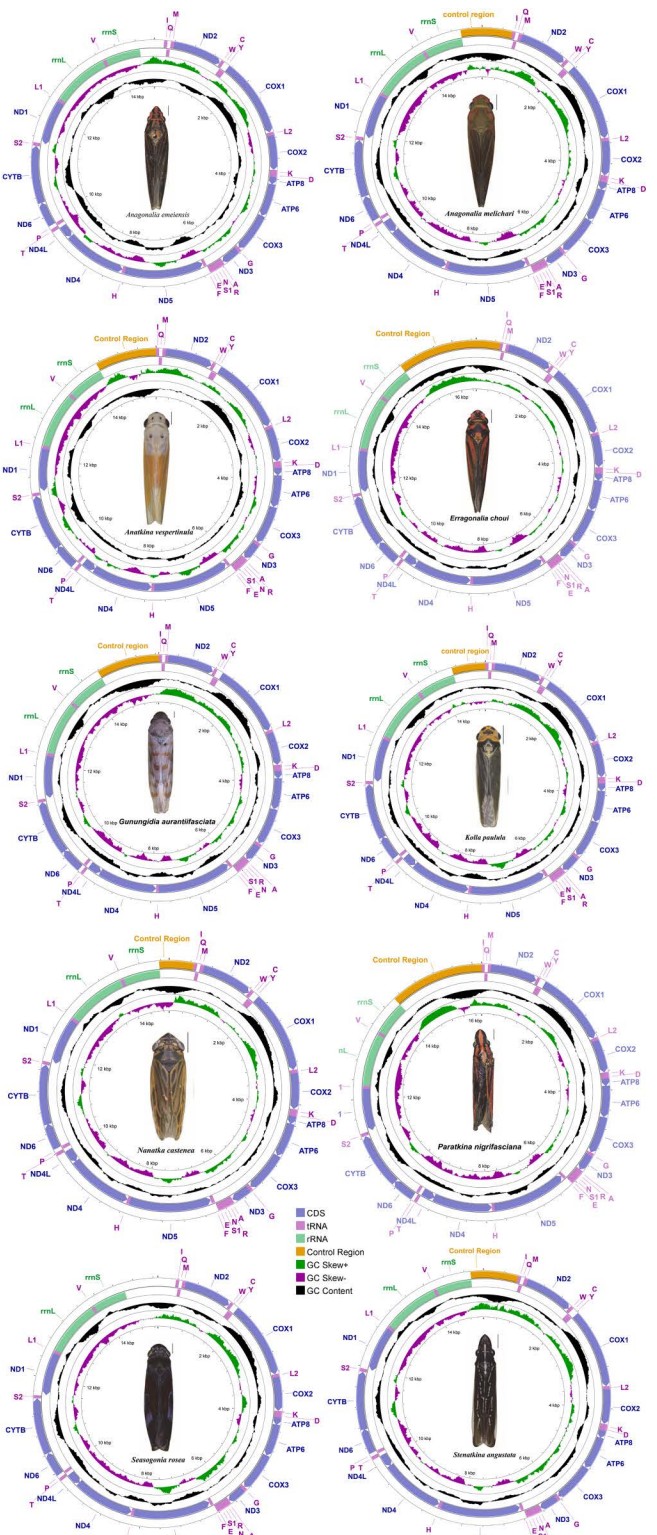

**Fig 1. The map illustrates the mitochondrial genome of ten Cicadellinae species.** Genes are depicted as blocks of different colors. Genes on the J-strand are indicated by color blocks outside the circle; genes on the N-strand are indicated by color blocks inside the circle.

The mitogenomes were significantly skewed toward A and T bases, with A+T content ranging from 76.1% (*K. paulula*) to 81.7% (*A. melichari*). The mitogenomes' PCGs (which ranged from −0.18 to −0.13) and rRNA (which ranged from −0.22 to −0.1) genes were all negatively skewed, but the tRNA (which ranged from 0.011 to 0.028) genes were favorably skewed (S5 Table in S1 File). Different levels of genetic overlap and intergenic areas were observed in ten mitogenomes. The overlapping positions ranged from 12 to 16, and the spaced positions were from 4 to 10. The biggest overlapping region, measuring 10 bp, occurred where the sequence of trnS2 was included in that of nd1(*K. paulula*), while the longest intergenic spacer, measuring 15 bp, was discovered between trnS2 and nd1 (*S. rosea* and *A. vespertinula*).

## Protein-coding genes and codon usage

Of the 13 PCGs, the N chain contains *nd5, nd4, nd4l,* and *nd1*, whereas the J chain contains the other PCGs. All ten Cicadelinae atp8 start with a TTG codon, nd2 of *Anatkina vespertinula* starts with a GTG codon, and the rest of the PCGs have traditional ATN (ATT/ATA/ATC/ATG) start codons. Among the 13 PCGs of 10 Cicadellinae species, most ended with the complete stop codon TAA or TAG, except for *cox2* and *nd5*, which ended with an incomplete stop codon T (S6 Table in S1 File.

The 10 mitogenomes under analysis have PCGs with total lengths ranging from 10,931 bp to 10,979 bp. The relatively low A%+T% of 10 mitotic genomes ranged from 75% to 80.7%, while the AT skew only slightly varied across the mitogenomes (S5 Table in S1 File).

In the current Cicadellinae mitogenomes, the relative synonymous codon usage (RSCU) of PCGs were computed and shown. The findings indicate that UUA (Leu), UCA (Ser), and CGA (Arg) are the three codons that are most commonly used, whereas CUC/G (Leu), AGC (Ser), and GGC (Gly) are rarely used (Fig 2). Similar results appear in other leafhopper mitogenomes. Only serine (AGN/UCN) has two codons out of the 19 amino acids (S7 Table in S1 File).

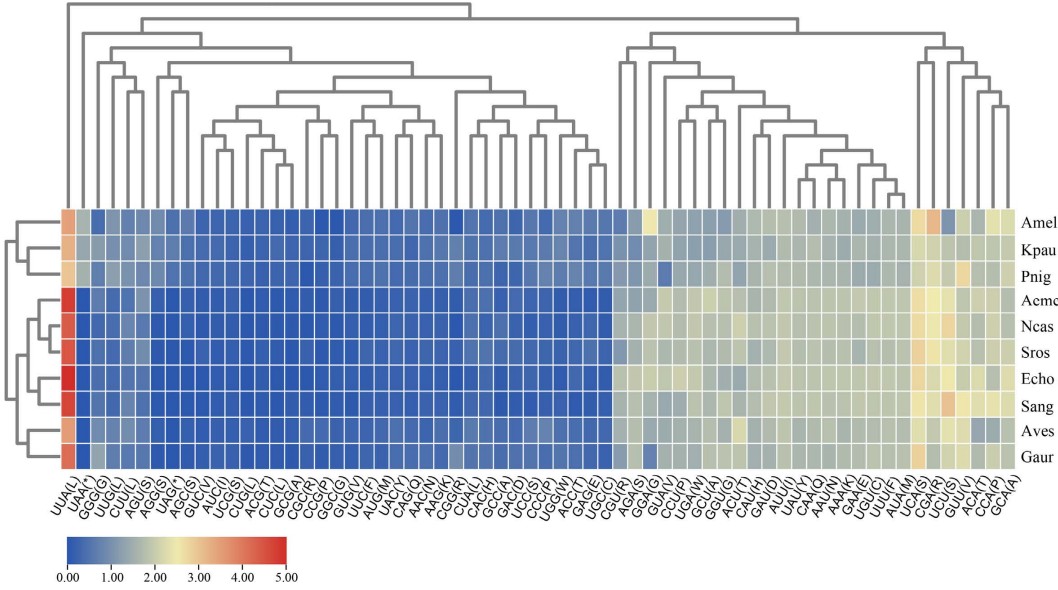

**Fig 2. Relative synonymous codon use (RSCU) of PCGs throughout the ten mitogenomes of Cicadellinae.** The hierarchical clustering of Cicadellinae species and codon frequencies is represented by the x- and y-axes, respectively. Aeme: *Anagonalia emeiensis,*Amel: *Anagonalia melichari*, Aves: *Anatkina vespertinula*, Echo: *Erragonalia choui*, Gaur: *Gunungidia aurantiifasciata*, Kpau: *Kolla paulula*, Ncas: *Nanatka castenea,* Pnig: *Paratkina nigrifasciana,* Sros: *Seasogonia rosea,* Sang: *Stenatkina angustata.*

For every PCG of the 22 leafhoppers in the subfamily Cicadellinae, the nonsynonymous substitutions (Ka), synonymous substitutions (Ks), and Ka/Ks(ω) values were computed. The specifics are displayed in Fig 3 and S8 Table in S1 File. With all 13 PCGs under purifying selection, their Ka/Ks values (which varied from 0.147 for *cox1* to 0.738 for *atp8*) were less than 1. The 13 PCGs' rates of evolution rose in the following order: *cox1 < cox3 < cytb < cox2 < nd1 < nd3 < atp6 < nd6 < nd4l < nd5 < nd2 < nd4 < atp8*.

## Transfer and ribosomal RNA genes

Across 10 sequenced mitogenomes, the overall length of rRNAs varied from 1900 to 2011 bp, while the AT contents varied from 79.6 to 83.1%, with a positive GC skew and a negative AT skew.

Of the 22 tRNA genes, 8 had a positive GC skew (0.108–0.204) and a positive AT skew (0.011–0.028), with 14 of the genes encoded by the J-strand and the remaining eight were located on the N-strand. All tRNA genes were able to fold into the typical cloverleaf secondary structure, exception of trnS1, which had a simple loop in place of a stable dihydrouridine (DHU) stem (Figs S1–S10 in S1 File).

## Control region

The control region, sometimes referred to as the A+T-rich region, is the biggest noncoding region in the mitogenomes and comprises the replication and transcription origins [22]. The putative control region ranged in length, ranging from 486 bp (*A. emeiensis*) to 2195 bp (*E. choui*), and its AT contents varied from 77.2% (*K. paulula*) to 92% (*N. castenea*), and it was located between the rrnS and trnI genes. Except for *A. emeiensis, E. choui* and *P. nigrifasciana,* which had positive AT and GC skews, the remaining *A. melichari* and *N. castenea* had negative AT and GC skews, respectively (S5 Table in S1 File).

## Phylogenetic relationship

For phylogenetic analyses of three datasets (AAs, PCGs, and PCGs+rRNA) of 24 species from the subfamily Cicadellinae and two outgroups (S9 Table in S1 File). [6,50], six phylogenetic trees were created using ML and BI methods.

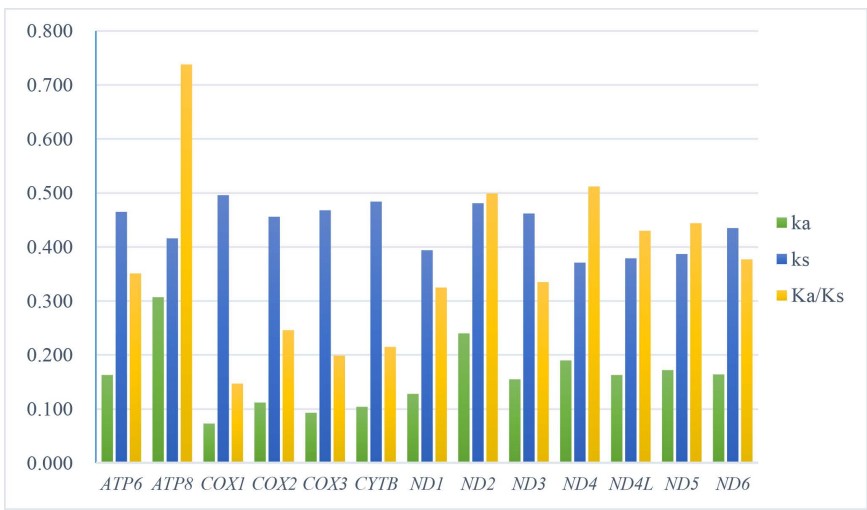

**Fig 3. The study examined the evolutionary rate of 13 PCGs in the mitogenomes of 22 leafhoppers.** The rate of nonsynonymous substitutions to synonymous substitutions is expressed as Ka/Ks, where Ks represents synonymous nucleotide substitutions per synonymous site and Ka is the nonsynonymous nucleotide substitutions per nonsynonymous site.

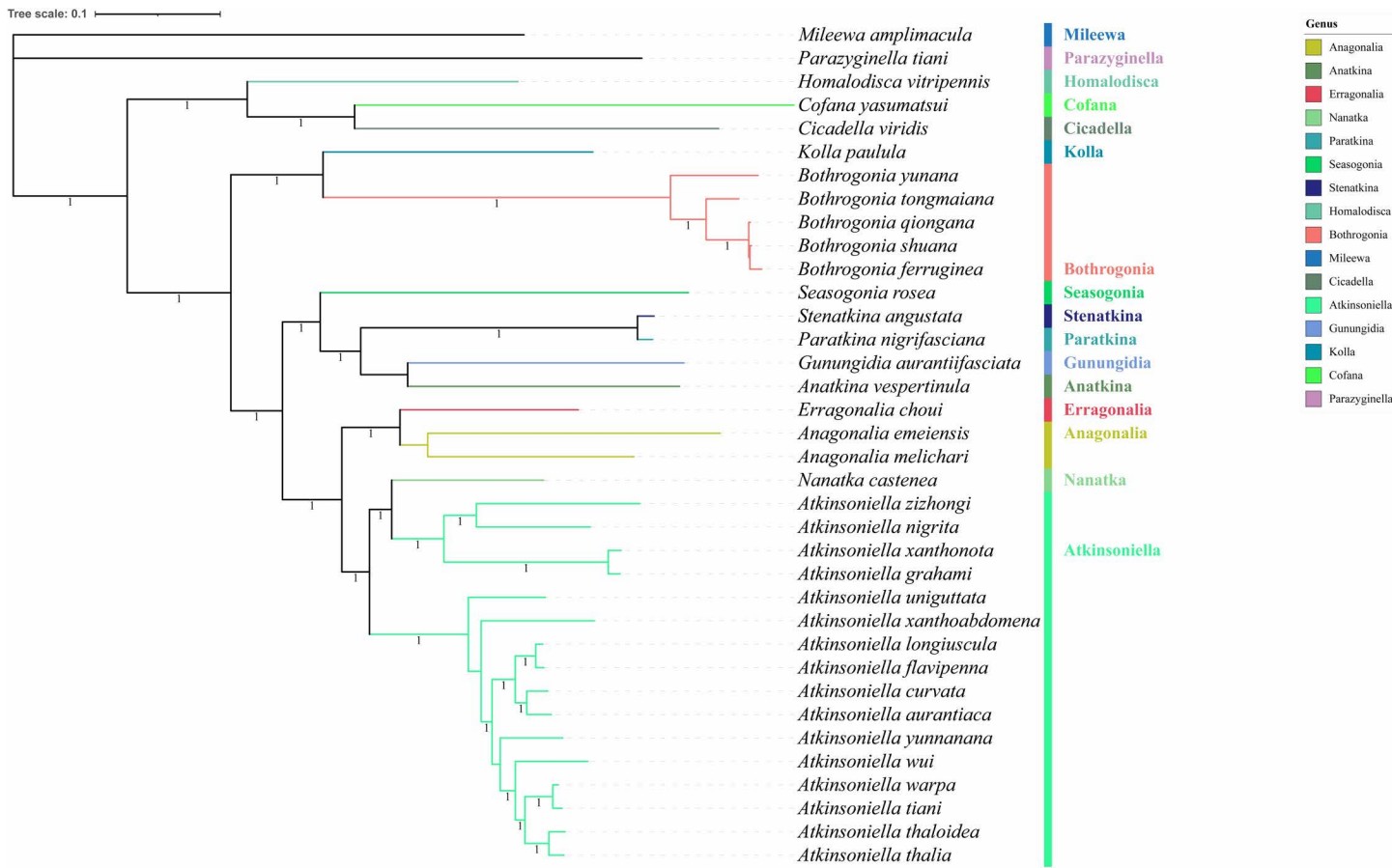

**Fig 4. Based on amino acids (AAs), phylogenetic trees generated by Bayesian inference (BI).** Branches display the bootstrap percentage (BP) and Bayesian posterior probabilities (BPPs).

PartitionFinder chose the models and partitioning schemes that worked best (S10 Table in S1 File). Phylogenetic studies showed that the topological structure calculated by various methods for various datasets was almost identical (Figs 4 and 5 and FigsS11-S14 in S1 File).

The topological structures of all six phylogenetic trees were completely consistent, and the majority of nodes have high support values. The genera *Atkinsoniella* and *Bothrogonia* remained monophyletic, and the relationship between genera within Cicadellinae consistently showed a pattern of (((((Atkinsoniella + Nanatka) + (Anagonalia + Erragonalia)) + (((Anatkina + Gunungidia) + (Paratkina + Stenatkina)) + Seasogonia)) + (Bothrogonia + Kolla)) + ((Cicadella + Cofana) + Homalodisca)) across all phylogenetic trees.

## Conclusion

The current research focused on the sequencing and analysis of ten leafhopper mitogenomes that belong to the Cicadellinae subfamily. In accordance with earlier studies [51,52], it was determined that the gene orders exhibited were consistent with the standard gene arrangement observed in insects. The length of the ten mitogenomes fell within 14,768 bp in *A. emeiensis* to 16,194 bp in *P. nigrifasciana*. The primary factor contributing to the variation in length is the discrepancy in intergenic spacer regions and the control region's length. In all ten mitogenomes, a 2 bp intergenic spacer region was found

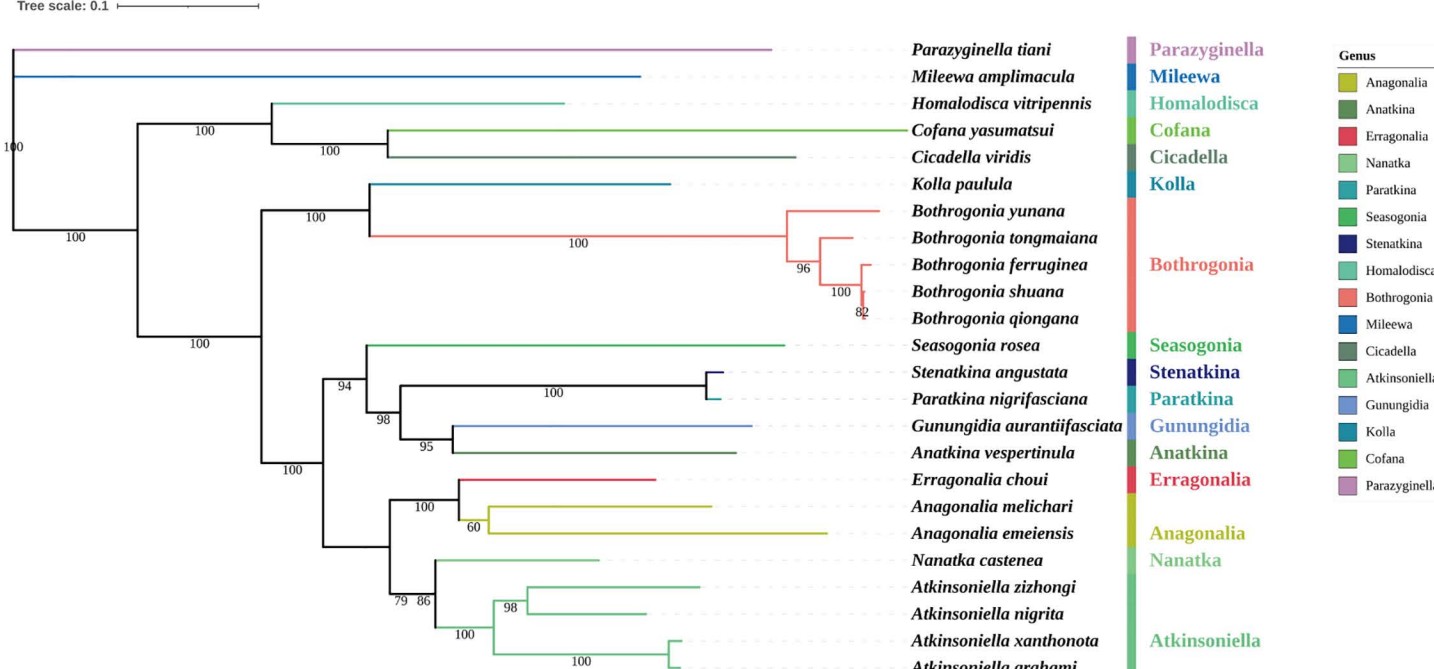

**Fig 5. Based on amino acids (AAs), phylogenetic trees generated by maximum likelihood (ML).** Branches display the bootstrap percentage (bp).

in both nd2-trnW and trnT-nad4L. In *A. melichari,* a 15 bp intergenic spacer was located between trnW and trnC; this spacer was not present in the mitogenomes of other Cicadellinae species. The PCGs in the ten mitogenomes that were recently sequenced displayed a length disparity of no more than 30 bp, implying that these genes possess a relatively stable set of features across different species. Because of the stability of their secondary structures, the length changes of this tRNA and rRNA in various species were restricted. Of all the tRNA genes, the trna S1 of ten species of Cicadellinae did not have a stable dihydrouridine (DHU) stem [53–59]. Within the Cicadellinae subfamily, TAG or a solitary T was the least frequently employed stop codon, while the predominant stop codon utilized by most protein-coding genes (PCGs) was TAA [60].

In this study, we successfully determined and analyzed the mitogenomes of ten species. Six phylogenetic trees were generated, each clearly depicting the relationships among Cicadellinae.. *E. choui* and *Anagonalia* are sister groups, and *Atkinsoniella* and *Nanatka* are always grouped together in a single clade, which was discovered by deconstructing the male genitalia because *E. choui* and *Anagonalia* do not have the important structural feature of aedeagus paraphysis, and they are distinguished from the large branch of ((*Anatkina* + *Gunungidia*) + (*Paratkina* + *Stenatkina*)) + *Seasogonia* by the presence or absence of lamellar or dentate projections of the aedeagus shaft and by the fact that the connective is "Y" and "V" shaped, respectively. The current lack of Cicadellinae mitogenomes hinders the ability to study the subfamily's phylogeny. Obtaining additional molecular data from Cicadellinae species in various geographic regions is necessary to better understand the phylogenetic relationships of both Cicadellinae and higher taxa.

## Supporting information

**S1 File.** S1 Fig. Predicted secondary cloverleaf structure for the tRNAs of *Anagonalia emeiensis.* S2 Fig. Predicted secondary cloverleaf structure for the tRNAs of *Anagonalia melichari*. S3 Fig. Predicted secondary cloverleaf structure for the tRNAs of *Anatkina vespertinula.* S4 Fig. Predicted secondary cloverleaf structure for the tRNAs of *Erragonalia choui.*

S5 Fig. Predicted secondary cloverleaf structure for the tRNAs of *Gunungidia aurantiifasciata.* S6 Fig. Predicted secondary cloverleaf structure for the tRNAs of *kolla paulula.* S7 Fig. Predicted secondary cloverleaf structure for the tRNAs of *Nanatka castenea*. S8 Fig. Predicted secondary cloverleaf structure for the tRNAs of *Paratkina nigrifasciana.* S9 Fig. Predicted secondary cloverleaf structure for the tRNAs of *Seasogonia rosea.* S10 Fig. Predicted secondary cloverleaf structure for the tRNAs of *Stenatkina angustata*. S11 Fig. Phylogenetic trees inferred by Bayesian inference (BI) based on the 13 protein-coding genes (PCGs). Bayesian posterior probabilities (BPPs) and bootstrap percentages (BP) are indicated on branches. S12 Fig. Phylogenetic trees inferred by Bayesian inference(BI) based on the 13 protein-coding genes and two rRNA genes (PCGs + rRNA). Bayesian posterior probabilities (BPPs) and bootstrap percentages (BP) are indicated. S13 Fig. Phylogenetic trees inferred by maximum likelihood (ML) based on the 13 protein-coding genes (PCGs). Bootstrap percentage (bp) is indicated on branches. S14 Fig. Phylogenetic trees inferred by maximum likelihood (ML) based on the 13 protein-coding genes and two rRNA genes (PCGs + rRNA). Bootstrap percentage (bp) is indicated on branches. S1 Table. Collection information for the 10 Cicadellidae species in this study. S2 Table. Summary statistics of the sequenced species. S3 Table. Sequence read archive accessions. S4 Table. *Anagonalia emeiensis, Anagonalia melichari, Anatkina vespertinula, Erragonalia choui, Gunungidia aurantiifasciata, Kolla paulula, Nanatka castenea, Paratkina nigrifasciana, Seasogonia rosea, Stenatkina angustata*. S5 Table. Nucleotide composition of subfamily Cicadellinae mitochondrial. S6 Table. Start and stop codons of the mitochondrial genomes of subfamily Cicadellinae. S7 Table.Codon number and RSCU in the subfamily Cicadellinae species mitochondrial PCGs. S8 Table. The evolutionary rate of each PCGs in the mitogenomes of Cicadellinae. S9 Table. Mitochondrial genomes used for the phylogenetic analyses in this study. S10 Table. Best partitioning schemes and models based on different datasets for phylogenetic analysis.
(ZIP)

## Acknowledgments

The authors acknowledge the assistance with software analysis provided by Feng Zhang of Nanjing Agricultural University, Nanjing, China, and the specimen collection provided by Wang Xianyi, Yang Lu, Li Fenge, Yu Xiaofei, Wu Qichao, Yu Zhou, and Xu Xiaoli.

## Author contributions

**Conceptualization:** Renhuai Dai, Maofa Yang.

**Data curation:** Likun Zhong.

**Formal analysis:** Likun Zhong, Yan Jiang, Bin Yan, Renhuai Dai.

**Funding acquisition:** Xiaofei Yu, Maofa Yang.

**Investigation:** Likun Zhong, Yan Jiang, Xiaofei Yu, Bin Yan.

**Methodology:** Yan Jiang, Bin Yan, Maofa Yang.

**Project administration:** Xiaofei Yu.

**Resources:** Likun Zhong, Xiaofei Yu, Renhuai Dai, Maofa Yang.

**Software:** Likun Zhong, Bin Yan.

**Supervision:** Maofa Yang.

**Validation:** Yan Jiang.

**Visualization:** Likun Zhong.

**Writing – original draft:** Likun Zhong.

**Writing – review & editing:** Renhuai Dai, Maofa Yang.

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
