## [Decision Letter · Decision Letter 0]

7 May 2024

Dear Dr. Yang,

Thank you for submitting your manuscript to PLOS ONE. After careful consideration, we feel that it has merit but does not fully meet PLOS ONE’s publication criteria as it currently stands. Therefore, we invite you to submit a revised version of the manuscript that addresses the points raised during the review process.

We look forward to receiving your revised manuscript.

Kind regards,

James Lee Crainey, Ph.D.

Academic Editor

PLOS ONE

10.7717/peerj.14026

In your revision ensure you cite all your sources (including your own works), and quote or rephrase any duplicated text outside the methods section. Further consideration is dependent on these concerns being addressed.

“The Guizhou Province Science and Technology Innovation Talent Team Project (grant No. Qian Ke He Pingtai Rencai–CXTD [2021]004); the National Natural Science Foundation of China (32360393).”

5. We notice that your supplementary figures are uploaded with the file type 'Figure'. Please amend the file type to 'Supporting Information'. Please ensure that each Supporting Information file has a legend listed in the manuscript after the references list.

Reviewers' comments:

Reviewer's Responses to Questions

**Comments to the Author**

1. Is the manuscript technically sound, and do the data support the conclusions?

Reviewer #1: Yes

Reviewer #2: Yes

2. Has the statistical analysis been performed appropriately and rigorously?

Reviewer #1: N/A

Reviewer #2: N/A

3. Have the authors made all data underlying the findings in their manuscript fully available?

Reviewer #1: No

Reviewer #2: Yes

4. Is the manuscript presented in an intelligible fashion and written in standard English?

Reviewer #1: Yes

Reviewer #2: Yes

Reviewer #1: The manuscript entitled "Phylogenetic investigation and mitochondrial genome description of ten species in nine genera of Cicadellinae from China (Hemiptera: Cicadellidae) " reports the sequencing, assembly, annotation, and phylogenetic analysis of ten mitochondrial genomes from nine genera of Cicadellinae. This work provides a valuable genetic resource for this subfamily. Key results include the identification of gene features across different species, variations in transfer and ribosomal RNA genes, and insights regarding the phylogeny within the Cicadellinae subfamily.

Overall, the MS demonstrates a scientifically sound approach to studying the mitochondrial genomes and phylogenetic relationships of Cicadellinae species. The methodology applied to assemble and analyze the genomes was well described. However, there are some issues regarding methods and discussion, as commented below.

1. The authors may provide a supplementary table with summary statistics of the sequenced species, for example, number of sequenced raw reads, number of clean reads, number of reads used to assemble the mitochondrial genomes, mapping information, such as coverage and depth of the assembled genomes.

2. The raw reads must be deposited to a public repository (for example, SRA) and made available.

3. There is limited explanation of the methodology used for phylogenetic analyses. On Materials and Methods, section 2.3 Phylogenetic analyses, the authors stated that “the best partitioning scheme and evolutionary models for 13 predefined partitions were selected using PartitionFinder2”, however the IQ-TREE (ML analysis) was used on ‘Auto’ mode for model selection. If IQ-TREE automatically selected the model, it is unclear why the best partition scheme and model were selected with PartitionFinder2?

4. The authors claimed that all ten species of Cicadellinae had their whole mitochondrial genomes assembled, however on Fig. 2 there is no annotation of ‘Control Region’ for two species (Anagonalia emeiesis and Seasogonia rosea). Please correct the figure or provide clarification on the annotation process.

5. The authors stated at the introduction that the “leafhoppers are effective vectors of plant pathogens such as phytoplasmas, as well as other bacteria and viruses that cause illness in a variety of plants”. The MS lacks detailed discussion on the implications of the findings for broader evolutionary and ecological contexts. In the conclusion section, there is a lot of repeated information from the results.

Minor:

• The authors should improve figure captions with more detailed information about the figures themselves and methods employed to generate them. For example, what represents the inner circles of Fig. 1?

• The authors should revise the language. For example, “topological topologies”.

Reviewer #2: The results presented in this work are robust and well-described. However, to ensure the reproducibility of the study, it is crucial to include detailed information about the library preparation kits and sequencing method used, as well as the sequencing depth (number of sequenced reads) for each sample. I recommend that this information be provided as supplementary material, which will significantly enhance the contribution of your article to the scientific community.

**Do you want your identity to be public for this peer review?** For information about this choice, including consent withdrawal, please see our Privacy Policy

Reviewer #1: **Yes: ** Daniel A. Moreira

Reviewer #2: No

---

## [Author Response · Author response to Decision Letter 1]

23 Jan 2025

Dear Reviewer,

Greetings from all authors!

We express our gratitude for your meticulous perusal, helpful comments, and constructive suggestions, which have substantially enhanced the exposition of our manuscript. We have carefully considered all comments and revised our manuscript accordingly, which includes writing polish, data availability, etc. For more details about our revised work, kindly refer to the response provided below.

Query: The authors may provide a supplementary table with summary statistics of the sequenced species, for example, number of sequenced raw reads, number of clean reads, number of reads used to assemble the mitochondrial genomes, mapping information, such as coverage and depth of the assembled genomes.

Response: Thanks for your suggestions. Table has been supplemented (Table S2).

Q: The raw reads must be deposited to a public repository (for example, SRA) and made available.

Response: Regards for your recommendations. Raw data has been uploaded to the Sequence Read Archive (Table S3).

Q: There is limited explanation of the methodology used for phylogenetic analyses. On Materials and Methods, section 2.3 Phylogenetic analyses, the authors stated that “the best partitioning scheme and evolutionary models for 13 predefined partitions were selected using PartitionFinder2”, however the IQ-TREE (ML analysis) was used on ‘Auto’ mode for model selection. If IQ-TREE automatically selected the model, it is unclear why the best partition scheme and model were selected with PartitionFinder2?

Response: Thank for your comment. IQ-TREE (ML analysis) was used on 'Auto' mode for model selection in order to identify PartitionFinder2 selected partitions and models automatically (Table S8). The description in the text has been changed (Line number: 120).

Q: The authors claimed that all ten species of Cicadellinae had their whole mitochondrial genomes assembled, however on Fig. 2 there is no annotation of ‘Control Region’ for two species (Anagonalia emeiesis and Seasogonia rosea). Please correct the figure or provide clarification on the annotation process.

Response: Thank you for your comments, (Anagonalia emeiesis and Seasogonia rosea) the control area is incomplete and we have used Geneious for mapping and have never been able to complete the identification of the full control area, but its completeness does not affect the phylogenetic study, and the terminology has now been changed for the 10 species of COMPLETE (Line number: 145, 259).

Q: The authors stated at the introduction that the “leafhoppers are effective vectors of plant pathogens such as phytoplasmas, as well as other bacteria and viruses that cause illness in a variety of plants”. The MS lacks detailed discussion on the implications of the findings for broader evolutionary and ecological contexts. In the conclusion section, there is a lot of repeated information from the results.

Response: Thank you for your valuable comments. Leafhoppers cause direct feeding damage to plants and transmit plant pathogens. Some leafhopper species are agricultural pests, but this was not the focus of the Cicadelliane phylogenetic study in this paper, so they are not discussed in their ecological context. Duplicate information in the conclusion has been changed.

Q: The authors should improve figure captions with more detailed information about the figures themselves and methods employed to generate them. For example, what represents the inner circles of Fig. 1?

Response: Thanks for your comments. Genes are depicted as blocks of different colors. Genes on the J-strand are indicated by color blocks outside the circle; genes on the N-strand are indicated by color blocks inside the circle. Each circle is coloured in a different way to represent different information within the mitochondrial genome, such as the location and orientation of genes, the amount of GC, etc. . For example, the inner circle of Figure 1 represents the size range of the mitochondrial genome (Line number: 147).

Reviewer 2

Query: The results presented in this work are robust and well-described. However, to ensure the reproducibility of the study, it is crucial to include detailed information about the library preparation kits and sequencing method used, as well as the sequencing depth (number of sequenced reads) for each sample. I recommend that this information be provided as supplementary material, which will significantly enhance the contribution of your article to the scientific community.

Response: Thanks for your suggestions. The library preparation kits and sequencing methods used are described in Materials and Methods, and the kits were used according to the instructions, as well as the depth of sequencing (number of sequencing reads) for each sample, which is provided as Supplementary Material (Table S2).

---

## [Decision Letter · Decision Letter 1]

11 Feb 2025

Dear Dr. Yang,

Thank you for submitting your manuscript to PLOS ONE. After careful consideration, we feel that it has merit but does not fully meet PLOS ONE’s publication criteria as it currently stands. Therefore, we invite you to submit a revised version of the manuscript that addresses the points raised during the review process.

We look forward to receiving your revised manuscript.

Kind regards,

James Lee Crainey, Ph.D.

Academic Editor

PLOS ONE

**Journal Requirements:**

Reviewers' comments:

Reviewer's Responses to Questions

**Comments to the Author**

Reviewer #1: All comments have been addressed

Reviewer #2: All comments have been addressed

2. Is the manuscript technically sound, and do the data support the conclusions?

Reviewer #1: Yes

Reviewer #2: Yes

3. Has the statistical analysis been performed appropriately and rigorously?

Reviewer #1: N/A

Reviewer #2: N/A

4. Have the authors made all data underlying the findings in their manuscript fully available?

Reviewer #1: Yes

Reviewer #2: Yes

5. Is the manuscript presented in an intelligible fashion and written in standard English?

Reviewer #1: Yes

Reviewer #2: Yes

**Reviewer #1: ** (No Response)

**Reviewer #2:**  The authors should clarify the discrepancy regarding the sequencing platform. In the manuscript, they report using the Illumina NovaSeq 6000, whereas in the NCBI/SRA submission, the sequencing platform is listed as the Illumina HiSeq 2000. This inconsistency should be addressed to ensure data accuracy and reproducibility.

**Do you want your identity to be public for this peer review?** For information about this choice, including consent withdrawal, please see our Privacy Policy

Reviewer #1: **Yes: ** Daniel Andrade Moreira

Reviewer #2: No

---

## [Author Response · Author response to Decision Letter 2]

26 Mar 2025

Dear Reviewer,

We express our gratitude for your meticulous perusal, helpful comments, and constructive suggestions, which have substantially enhanced the exposition of our manuscript.

Query: The authors should clarify the discrepancy regarding the sequencing platform. In the manuscript, they report using the Illumina NovaSeq 6000, whereas in the NCBI/SRA submission, the sequencing platform is listed as the Illumina HiSeq 2000. This inconsistency should be addressed to ensure data accuracy and reproducibility.

Response: We sincerely appreciate the reviewer’s attention to detail. The inconsistency between the manuscript and the NCBI/SRA submission arose due to an inadvertent error during metadata entry in the SRA submission portal. To resolve this: Clarification of Sequencing Platform: The sequencing was performed using the Illumina NovaSeq 6000 platform, as stated in the manuscript. The reference to the HiSeq 2000 in the SRA metadata was incorrect and has now been updated in the NCBI/SRA records (BioProject ID: [PRJNA1200972, PRJNA1211036]).

We regret this oversight and thank the reviewer for highlighting it. All analyses and conclusions remain unaffected, as the sequencing platform does not alter the downstream bioinformatics workflows employed.

Sincerely,

Maofa Yang

---

## [Editor Report · Decision Letter 2]

23 Jul 2025

Phylogenetic investigation and mitochondrial genome description of ten species in nine genera of Cicadellinae from China (Hemiptera: Cicadellidae)

PONE-D-24-10527R2

Dear Dr. Yang,

We’re pleased to inform you that your manuscript has been judged scientifically suitable for publication and will be formally accepted for publication once it meets all outstanding technical requirements.

Kind regards,

James Lee Crainey, Ph.D.

Academic Editor

PLOS ONE
---

## [Editor Report · Acceptance letter]

PONE-D-24-10527R2

PLOS ONE

Dear Dr. Yang,

I'm pleased to inform you that your manuscript has been deemed suitable for publication in PLOS ONE. Congratulations! Your manuscript is now being handed over to our production team.

Kind regards,

on behalf of

Dr. James Lee Crainey

Academic Editor

PLOS ONE